# Optimal-state Dynamics Estimation for Physics-based Human Motion Capture from Videos

**Cuong Le**[1], **Viktor Johansson**[1], **Manon Kok**[2] and **Bastian Wandt**[1]

[1]Department of Electrical Engineering, Linköping University, Sweden
[2]Delft Center for Systems and Control, Delft University of Technology, The Netherlands

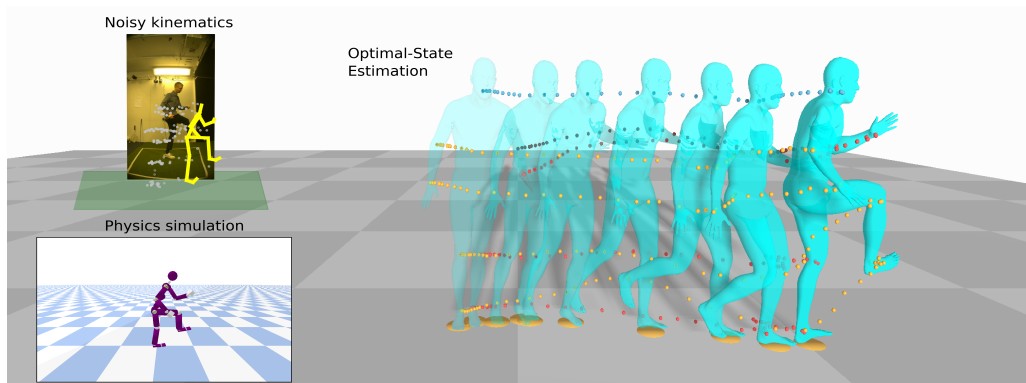

Figure 1: *OSDCap* is an optimal-state dynamics estimation (cyan) based on two streams of input motion, a kinematics-based pose estimation from videos (top-left), and a physics-based simulation by a meta-PD controller (bottom-left). The predicted motion is physically-plausible, contains reduced high-frequency noise, while retaining highly accurate global position.

## Abstract

Human motion capture from monocular videos has made significant progress in recent years. However, modern approaches often produce temporal artifacts, e.g. in form of jittery motion and struggle to achieve smooth and physically plausible motions. Explicitly integrating physics, in form of internal forces and exterior torques, helps alleviating these artifacts. Current state-of-the-art approaches make use of an automatic PD controller to predict torques and reaction forces in order to re-simulate the input kinematics, i.e. the joint angles of a predefined skeleton. However, due to imperfect physical models, these methods often require simplifying assumptions and extensive preprocessing of the input kinematics to achieve good performance. To this end, we propose a novel method to selectively incorporate the physics models with the kinematics observations in an online setting, inspired by a neural Kalman-filtering approach. We develop a control loop as a meta-PD controller to predict internal joint torques and external reaction forces, followed by a physics-based motion simulation. A recurrent neural network is introduced to realize a Kalman filter that attentively balances the kinematics input and simulated motion, resulting in an optimal-state dynamics prediction. We show that this filtering step is crucial to provide an online supervision that helps balancing the shortcoming of the respective input motions, thus being important for not only capturing accurate global motion trajectories but also producing physically plausible human poses. The proposed approach excels in the physics-based human pose estimation task and demonstrates the physical plausibility of the predictive dynamics, compared to state of the art. The code is available on ⓞ.

38th Conference on Neural Information Processing Systems (NeurIPS 2024).

# 1   Introduction

Three-dimensional human motion estimation is a long-standing and challenging research goal in computer vision, particularly in monocular scenarios due to inherent depth ambiguities. Previous approaches have incorporated kinematic priors, e.g. by enforcing smoothness, maintaining bone length constancy, or imposing symmetry constraints. However, due to inconsistencies of frame-wise predictions, these solutions do not necessarily lead to physically plausible motions. This has led to the emergence of a new research direction that combines traditional 3D motion estimation with physical models of the human skeleton. Instead of directly predicting a human pose, these approaches estimate the internal joint torques and exterior forces that drive the motion. Consequently, physics simulators are employed to obtain the resulting motion [37, 38, 8, 50, 21].

However, since simulators are never perfect representations of the real world, they introduce inevitable errors, where the complex human body was never fully modelled, only approximation by rigid body dynamics [3]. Moreover, measurements, including "ground truth" recordings, are inherently noisy. To tackle these problems, we propose *OSDCap*, a state-aware architecture that combines a differentiable physical simulation with our novel neural Kalman filtering approach. Fig. 1 shows our reconstructed poses predicted from noisy kinematics estimates as well as the estimated dynamics from a video.

*OSDCap* is an online filtering and dynamics estimation that can be trained in an end-to-end manner. In detail, our approach consists of two steps, starting from a noisy kinematics reconstruction obtained by an off-the-shelf video-based 3D human pose estimator: 1) a simulation branch that estimates joint torques using a PD controller and computes the resulting motion, 2) an adaptive filtering stage that combines the output of the simulation stage and the video-based kinematics input to produce a refined motion. We follow prior work that utilizes the meta-PD algorithm for torque calculation [38, 21] to simulate plausible motion. However, the effectiveness of the PD algorithm heavily relies on the choice of the P and D gains [38] and on an accurate model of the human kinematic chain, which is generally unknown. Moreover, the measurements from the monocular 3D kinematics pose estimator contain a large amount of noise, which ultimately leads to inaccurate predictions. Shimada et al. [38] mitigate these problem by introducing an additional offset term into the PD controller. While this approach still produces reasonable output motion it is neither physically explainable nor consistent with PD controllers in control theory. We aim to solve this problem at the root by taking inspiration from control theory and propose a solution for processing the imperfect PD calculation by a learnable Kalman filtering method [36]. The proposed Kalman filter takes the simulated motions and the noisy 3D pose estimation as inputs, combines them, and produces an optimal state prediction as the output. The Kalman filter effectively refines the PD controller-based simulated motions into more plausible and realistic motions. While the Kalman filter fixes inaccurate kinematic measurements from the 3D pose estimator it does not take different weight distributions in the human body parts into account. We calculate an initial weight distribution – the inertia matrix – for an average human body shape. However, as for the skeletal structure, these are only approximations that lead to inaccurate simulations. We mitigate this issue by predicting an inertia bias matrix in each time step which is added to the initial inertia matrix.

We demonstrated the 3D reconstruction performance of our method on the popular Human3.6M [15] dataset, and the newer Fit3D [7] and SportsPose [14] datasets, comparing them with recent state-of-the-art physics-based methods.

In summary, *OSDCap* introduces a new physics-based human motion and dynamics estimation method leveraging a learnable Kalman filter and a learnable inertia prediction, that produces plausible motion as well as valuable estimates of exterior forces and internal torques. By offering improved accuracy and interpretability in human motion estimation, *OSDCap* presents a promising step towards bridging the gap between computer vision and the complex physics-based human motion modeling.

# 2   Related Work

## 2.1   Kinematics 3D Human Motion Capture

Monocular 3D human motion capture is a well-studied line of research, with common approaches that can be roughly divided into two groups, 1) end-to-end approaches that directly predict human poses from images [39, 29], and 2) lifting from 2D [1, 26, 28, 12, 4, 30, 42, 2, 10, 44, 22, 47, 43, 31]. Recent work addresses the problem by fitting volumetric models to 2D/3D evidence, aiming to achieve realistic human motion [27, 17, 25, 19, 20, 48, 45, 18, 23, 53, 40]. Despite the significant

progress, vision-based human 3D pose estimation is still an ill-posed problem, due to the loss of depth information from the monocular setup. Therefore, captured 3D motions often contain different types of implausibility, ranging from unnatural poses, jittering, or unrealistic body artifacts [46, 8].

## 2.2 Physics-based 3D Human Motion Capture

Recent studies [37, 50, 46, 8, 21] enforce physics as constraints for motion reconstruction, eliminating implausible artifacts created by the monocular estimation, i.e. jittering, ground penetration, and unnatural human poses.

Motion imitation using reinforcement learning (RL) is a popular approach for simulating physically plausible results [32, 49, 50, 33, 51]. RL-based methods enforce physics constraints in the reward functions, either from manually-designed formulas, or from physics engines. The bottleneck of RL-based approaches is the low transferability of the learned policies to unseen motions.

Motion optimization is another common approach for physics-based human motion capture. However, optimization problems often require a differentiable framework, thus, instead of relying on non-differentiable physics engines, prior studies [34, 37, 46] adapt simplified motion equations [5] as a dynamics constraint for simulated motions. More recent approaches [13, 9] manage to optimize through non-differentiable simulation using evolutionary optimization methods [11]. Gärtner et al. [8] implement a differentiable version of PyBullet [3], resulting in an optimizable framework with complex physics engines. However, most optimized motion solutions, similar to RL-based solutions, have limited adaptability to different data distributions, requiring re-optimizing on new sets of action.

Utilizing the generalizability of neural network models in an end-to-end manner is still an open line of research, due to the difficulty of finding physical plausibility patterns from data. Rempe et al. [35] utilizes a variational autoencoder architecture for predicting plausible motions, approximating the dynamics simulation by a decoder network. This assumption might result in unrealistic force prediction with respect to biomechanics literature. Li et al. [21] utilize the meta-PD controller with learnable parameters for torque prediction, but with an additional compensation term based on root residual forces. Zhang et al. [52] realize a transformer-based autoencoder to refine kinematics input sequences, while integrating physics constrain inside the latent embedding. However, both Li et al. [21] and Zhang et al. [52] make predictions based on the encoding of the full motion sample, i.e. they require knowledge of past and future motions, therefore, limiting the applicability of the method to offline setups, where future information is available. Shimada et al. [38] also use a meta-PD controller for calculating the optimal joint torques, which in turn generates a simulated motion matching visual kinematics estimation. Despite the plausibility of the estimated pose, the global precision of the motion in world coordinate is limited and not fully addressed.

We aim to leverage physics-based approach (with meta-PD controller) on motion data captured by monocular camera systems, in a recursive online setup, and expand the prediction to more complex practical movements such as sports.

## 3 Method

This section presents our proposed approach *OSDCap* in detail. We start by creating an average proxy character B based on the uniform human configuration from the Human3.6M dataset [15]. The character approximates a human body by circles and cylinders. Additionally, we leverage the pretrained neural network TRACE [40] to obtain an initial 3D pose estimation. Without any additional priors, *OSDCap* aims to predict the joint torques and external forces that drive the proxy character to match the kinematics evidence given by TRACE. Following prior work, we employ a neural network that predicts the parameters of a meta-PD controller which consecutively predicts the joint torques. While related approaches [38, 21] stop here, we note that the quality of the motion given by the PD controllers' prediction highly depends on the realism of the videos-based kinematic estimation and the proxy character. Since a model can always only be an approximation of the real world, this leads to inaccurate predictions which prior work compensates for by adding an additional offset term to the PD controller. Unfortunately, this not only introduces a non-physical assumption but through experimentation we found that this term attributes the major part to the prediction of the PD controller. We aim to maintain the physical plausibility of our approach by introducing a novel filtering approach inspired by a neural Kalman filter [36] to refine and update the motion states. Additionally, at each time step, the foot contact states and ground reaction forces are estimated directly from the motion leading to a full description of the system dynamics. Fig. 2 shows an overview of our method.

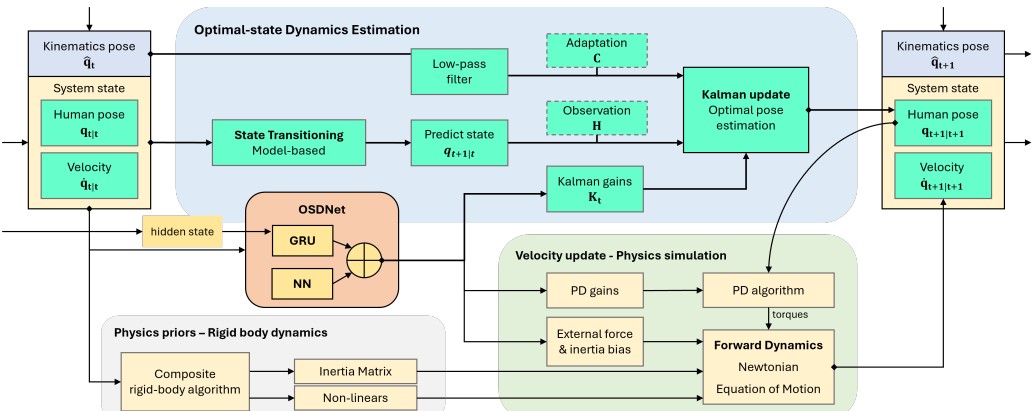

Figure 2: The main pipeline of *OSDCap*. Our approach consists of one neural network model, *OSDNet* (orange), and three processing components. *OSDNet* takes the current system state, estimates a Kalman gain matrix, PD gains, external force and an inertia-bias matrix. The optimal pose estimation performs contains a Kalman filter for the current system state and the input kinematics. Yellow refers to the algorithm's state vectors and cyan denotes processing operations. The physics priors block (gray) computes the inertia matrix and non-linear forces using the Composite rigid-body algorithm and Inverse dynamics [5]. Using the PD algorithm and forward dynamics (Eq. 1), the physics simulation block (green) updates the velocity based on the computed optimal pose and physics priors.

## 3.1 Preliminaries - Rigid Body Dynamics

Similar to previous studies [34, 37, 38, 46], we enforce physics constraints based on *Rigid Body Dynamics* [5], inline with the Newtonian equation of motion. For a total of $N$ keypoints, the full human pose is represented as a vector $\mathbf{q} \in \mathbb{R}^{6+3N}$, encoding the global translation and rotation in the first 6 entries, and internal joint angle states in the remaining $3N$ entries. $\dot{\mathbf{q}} \in \mathbb{R}^{6+3N}$ is the corresponding velocity vector. The motion dynamics of the captured human poses should satisfy the Newtonian equation of motion, expressed as

$$\mathbf{M}(\mathbf{q})\ddot{\mathbf{q}} = \boldsymbol{\tau} + \boldsymbol{\lambda} - \mathbf{h}(\mathbf{q}, \dot{\mathbf{q}}), \tag{1}$$

where $\mathbf{M}(q) \in \mathbb{R}^{(6+3N) \times (6+3N)}$ is the inertia matrix, computed from the proxy character, $\ddot{\mathbf{q}} \in \mathbb{R}^{6+3N}$ is the acceleration, $\boldsymbol{\tau} \in \mathbb{R}^{6+3N}$ are the internal joint torques, $\boldsymbol{\lambda} \in \mathbb{R}^{6+3N}$ are the external forces, and $\mathbf{h}(\mathbf{q}, \dot{\mathbf{q}}) \in \mathbb{R}^{6+3N}$ is the non-linear term including gravitational, Coriolis, and centrifugal forces, computed using inverse dynamics with zero acceleration on the proxy character [6]. Our goal is to estimate the two vectors $\boldsymbol{\tau}$ and $\boldsymbol{\lambda}$ that produce plausible motion dynamics.

## 3.2 Optimal-state Dynamics Capture

The proposed *OSDCap* consists of three main processing stages: an optimal pose estimation, a physics priors calculation, and a velocity update based on physics simulation. The optimal pose estimation phase is a filtering approach inspired by KalmanNet [36] that estimates an optimal output pose based on the current system state and video-based 3D kinematics inputs. The physics priors calculation computes the current inertia matrix and non-linear forces based on the proxy character from the current system state. The physics simulation phase computes the next velocity state using the PD algorithm and forward dynamics (Eq. 1) from the estimated optimal pose.

The required inputs for the optimal pose estimation and physics simulation are estimated by our neural network *OSDNet*. *OSDNet* consists of two modules, one predicts Kalman gains for the Kalman filtering, and the other predicts PD gains, external forces and inertia-bias for the physics simulation.

**Optimal Pose Estimation.** As shown in Fig. 2, the Kalman filtering block takes the current system state and the videos-based kinematics pose as inputs. Following traditional Kalman filters the next predict state is computed from the previous state. We define the state transitioning phase as

$$\mathbf{q}_{t+1|t} = \mathbf{q}_{t|t} + \dot{\mathbf{q}}_{t|t}\Delta t. \tag{2}$$

The predicted positional state $\mathbf{q}_{t+1|t}$ is the physics-constrained body pose. The observation matrix $\mathbf{H}$ (Fig. 2) maps the predicted states $\mathbf{q}_{t+1|t}$ to an observed simulated positional state. $\mathbf{C}$ is an adaptation matrix to reduce the gap between observed states from videos, and observed states from physics simulation. $\mathbf{H}$ and $\mathbf{C}$ are optimized along with *OSDNet* while training, but stay constant during inference. From the current system states, a Kalman state update process is performed as

$$\mathbf{q}_{t+1|t+1} = \mathbf{q}_{t+1|t} + \mathbf{K}_t(\mathbf{C}\hat{\mathbf{q}}_t - \mathbf{H}\mathbf{q}_{t+1|t}), \tag{3}$$

where $\mathbf{K}_t$ contains the estimated Kalman gains at step $t$ based on the current states and observations. Inspired by [36], the Kalman gains estimation module is implemented as *Gated Recurrent Units*, which have the ability to propagate the latent motion dynamics throughout the simulated motion via the hidden states of the GRU. The prediction of Kalman gains requires information about the the system's state dynamics [36], thus four additional dynamics features need to be feed into *OSDNet*'s GRU input, namely: *observation*, *innovation*, *forward evolution* and *forward update*. They are calculated as

$$\Delta\text{evolution} = \mathbf{q}_{t|t} - \mathbf{q}_{t-1|t-1}$$
$$\Delta\text{update} = \mathbf{q}_{t|t} - \mathbf{q}_{t|t-1}$$
$$\Delta\text{innovation} = \mathbf{C}\hat{\mathbf{q}}_{t+1} - \mathbf{H}\mathbf{q}_{t+1|t} \tag{4}$$
$$\Delta\text{observation} = \mathbf{C}\hat{\mathbf{q}}_{t+1} - \mathbf{q}_t.$$

We modify the original design from [36] due to the practical reasons of our system. In self-occluded scenarios, the noisy input $\hat{\mathbf{q}}_t$ often contains artifacts such as body deformation and they often last for a period of time (approx. 10 frames). The intermediate difference between $\hat{\mathbf{q}}_t$ and $\hat{\mathbf{q}}_{t-1}$ in the original design [36] is not strong enough to model those artifacts, because they could both contain the same incorrect kinematic estimation. We change the calculation of $\Delta\text{observation}$ as in Eq. 4 better deal with mis-detection cases that would cause large responses in $\Delta\text{observation}$. The pose $\mathbf{q}_{t|t}$ is the optimal state at step $t$ and inherits the global translation estimation from kinematics observations while retaining the physical plausibility of the human pose from the physics simulation.

**Physics Simulation.** The purpose of the physics simulation stage in Fig. 2 is to update the velocity $\dot{\mathbf{q}}_{t|t}$ that best describes the dynamics of the filtering process. Therefore, the estimated pose $\mathbf{q}_{t+1|t+1}$ can be used as the target signal for the PD algorithm, calculating the joint torque $\boldsymbol{\tau}_t$ that maps the predict pose $\mathbf{q}_{t+1|t}$ to the optimal pose $\mathbf{q}_{t+1|t+1}$. The joint torque is predicted by the PD algorithm

$$\boldsymbol{\tau}_t = \boldsymbol{\kappa}_P(\mathbf{q}_{t+1|t+1} - \mathbf{q}_{t+1|t}) + \boldsymbol{\kappa}_D\dot{\mathbf{q}}_{t|t}, \tag{5}$$

where $\boldsymbol{\kappa}_P$, $\boldsymbol{\kappa}_D$ are proportional and derivative gains respectively. Inspired by [38], the meta-PD controller was applied at this stage, where $\boldsymbol{\kappa}_P$, $\boldsymbol{\kappa}_D$ are learnable and estimated from *OSDNet*. By using the filtered optimal pose as the target, no unrealistic temporal filtering or optimization is needed to refine the noisy kinematics inputs.

Additionally, the external forces are also estimated by *OSDNet*, assuming the source of external forces comes only from contact points and is computed as

$$\boldsymbol{\lambda}_t = \sum_c^2 \mathbf{J}_t^c\boldsymbol{\rho}_t^c\mathbf{f}_t^c, \tag{6}$$

where $\mathbf{J}_t^c$ is Jacobian matrix that maps linear velocity at contact point $c$ to rotational velocity of every other joints, $\boldsymbol{\rho}_t^c$ and $\mathbf{f}_t^c$ are the contact probability and the linear force vector at contact $c$. The three vectors are separately estimated by *OSDNet*.

**Inertia Estimation.** Since we do not have access to the real bone length and mass distribution of the human, there exists a knowledge gap between simulated human character and the real human subject, the inertia tensor computed by the composite rigid-body algorithm is sub-optimal. *OSDNet* is designed to also estimate an inertia bias term $\mathbf{M}_t^b$ that reduces this knowledge gap. The required acceleration to drive the current simulated pose to the next states is calculated as in Eq. 7.

$$\ddot{\mathbf{q}}_t = (\mathbf{M}(\mathbf{q}_t)^{-1} + \mathbf{M}_t^b)(\boldsymbol{\tau}_t + \boldsymbol{\lambda}_t - \mathbf{h}(\mathbf{q}_t, \dot{\mathbf{q}}_t)), \tag{7}$$

To update the system state, finite interpolation is applied, using the newly calculated acceleration $\ddot{q}_t$. The update process is given by

$$\dot{\mathbf{q}}_{t+1|t+1} = \dot{\mathbf{q}}_{t|t} + \ddot{\mathbf{q}}_t\Delta t, \tag{8}$$

where $\dot{\mathbf{q}}_{t+1|t+1}$ is the updated system state that represents the current system dynamics, under physics constraints from gravity and contact forces. The system now proceeds back to the transitioning phase in Eq. 2, creating a closed loop process that works recursively.

### 3.3 Objective Losses

To reconstruct the optimal state, we define the overall objective loss $L$ as a weighted sum of multiple loss functions as

$$L = \frac{1}{T} \sum_t^T \left( \omega_1 L_t^{\mathbf{P}_{t+1|t+1}} + \omega_2 L_t^{\mathbf{q}_{t+1|t+1}} + \omega_3 L_t^{\mathbf{P}_{t+1|t}} + \omega_4 L_t^{\mathbf{q}_{t+1|t}} + \omega_5 L_t^c + L_t^{\text{reg}} \right), \qquad (9)$$

where $\omega_1 = 0.5, \omega_2 = 0.1, \omega_3 = 0.7, \omega_4 = 0.2, \omega_5 = 0.4$ are weighting factors. The optimal reconstruction losses $L_t^{\mathbf{q}_{t+1|t+1}}$ and $L_t^{\mathbf{P}_{t+1|t+1}}$ measure the L1 distance between the estimated optimal pose $\mathbf{q}_{t+1|t+1}$ and its corresponding 3D keypoints (obtained from forward kinematics) with the ground-truth poses $\mathbf{q}_{t+1}^{GT}$ and ground-truth 3D keypoints $\mathbf{p}_{t+1}^{GT}$. The supervision for predict pose $\mathbf{q}_{t+1|t}$ is carried out similarly, ensuring the correct behaviour of the physics simulation. $L_t^c$ is the contact loss, using Binary Cross Entropy measurement between the predicted contact probabilities $\rho_t^c$ of two feet with pseudo-ground-truth contact binary labels $\hat{\rho}_t^c$. We generate the ground truth contact labels for training based on the foot-ground distances of ground-truth 3D keypoints. The individual losses are computed as

$$L_t^{\mathbf{P}_{t+1|t+1}} = \sum^N \|\mathbf{p}_{t+1}^{GT} - \mathbf{p}_{t+1|t+1}\|, \quad L_t^{\mathbf{q}_{t+1|t+1}} = \sum^{6+3N} \|\mathbf{q}_{t+1}^{GT} - \mathbf{q}_{t+1|t+1}\|,$$

$$L_t^{\mathbf{P}_{t+1|t}} = \sum^N \|\mathbf{p}_{t+1}^{GT} - \mathbf{p}_{t+1|t}\|, \qquad L_t^{\mathbf{q}_{t+1|t}} = \sum^{6+3N} \|\mathbf{q}_{t+1}^{GT} - \mathbf{q}_{t+1|t}\|, \qquad (10)$$

$$L_t^c = -\sum_{c=1}^2 \hat{\rho}_t^c \log(\rho_t^c) + (1 - \hat{\rho}_t^c) \log(1 - \rho_t^c).$$

By re-introducing a part of the noisy kinematics measurements into the prediction, an additional regularization loss $L_t^{\text{reg}}$ is beneficial to ensure smoothness and plausibility of the output motions. The regularization consists of three objectives: 1) $L_t^{acc}$ is the acceleration loss, computed as the absolute difference between $\ddot{\mathbf{q}}_t$ and $\ddot{\mathbf{q}}_{t-1}$, 2) $L_t^{vel}$ is the velocity loss, measuring the distance between the first-order difference of ground-truth motion $\mathbf{q}_{t+1}^{GT}$ and of estimated optimal $\mathbf{q}_{t+1}$, and 3) The friction loss $L_t^{fric}$ encourages the feet to stay in the same position during ground contact. With the regulator weighting of $\omega_6 = 0.14, \omega_7 = 0.03, \omega_8 = 0.28$, $L_t^{\text{reg}}$ is expressed as

$$L_t^{\text{reg}} = \omega_6 L_t^{acc} + \omega_7 L_t^{vel} + \omega_8 L_t^{fric}$$

$$= \omega_6 \sum^{6+3N} \|\ddot{\mathbf{q}}_t - \ddot{\mathbf{q}}_{t-1}\| + \omega_7 \sum^{6+3N} \|\mathbf{q}_{t+1}^{GT} - \mathbf{q}_t^{GT}\| - \|(\mathbf{q}_{t+1} - \mathbf{q}_t)\| + \omega_8 \sum_{c=1}^2 \rho_t^c \|(\mathbf{p}_{t+1}^c - \mathbf{p}_t^c)\|. \qquad (11)$$

## 4 Experiments

### 4.1 Datasets

We evaluate our approach on two human motion benchmark datasets. The first and main dataset is the popular Human3.6M dataset [15]. The dataset contains indoor 3D human motion capture data, including 2D and 3D keypoints, skeleton joint angles, and videos. Seven actors perform 15 different actions. Following previous work [38, 21], the first five subjects (S1, S5, S6, S7, S8) are used for training, and the last two (S9, S11) for evaluation. According to [37], only actions that have foot-ground contacts were considered. Details about the selected sequences are found in the supplemental document C.

The second database is Fit3D [7]. Fit3D contains indoor motion capture data for a variety of exercises. We split the data by taking samples from the 6 actors (s03, s04, s05, s07, s08, s10) for training, and 2 actors (s09, s11) for evaluation, inspired by the setup from [37] on Human3.6M.

Since the scene setting from Human3.6M and Fit3D are very similar, we perform an additional evaluation on the new dataset SportsPose [14], which consists of video-based sport action sequences with corresponding ground truth 3D keypoints. We use this dataset to show out-of-domain performance, since the 3D kinematics estimator TRACE [40] has not been trained on it.

## 4.2 Implementation Setups

The initial motion observation is generated by TRACE [40]. As suggested by [38, 8], all extracted motions are down-sampled from 100Hz to 50Hz. The samples are aligned to the world origin in the first frame, then split into 100-frame sub-sequences to utilize batch training and evaluation. The proxy character is created with respect to the provided skeleton metadata in Human3.6M [15], including the mean bone lengths and joint angles configuration. The inertia matrix and bias force (including gravitational, Coriolis, and centrifugal forces) are calculated online using RBDL [6], based on the state of the proxy character.

We train *OSDNet* in an end-to-end procedure. *OSDNet* consists of three fully-connected layers, followed by six different heads for PD gains ($\kappa_P$, $\kappa_D$), inertia bias ($\mathbf{M}^b$), contact probability ($\rho^c$), linear external force from the ground ($\lambda$), and Jacobian matrix ($\mathbf{J}$). These six entries are responsible for the motion simulation phase, following Eq. 1 and 6. The GRU units in the proposed optimal-state prediction module (cf. Fig. 2) take current system states, additional dynamics features (Eq. 4), and its hidden state $\mathbf{h}_{\text{gru}}$ as inputs. The output is the Kalman gain-matrix for the Kalman update process. For a details descriptions of the *OSDNet*'s architecture, please refer to the supplementary document A.

*OSDNet* is trained for 15 epochs with a base learning rate of $5e^{-4}$ and a batch size of 64. The learning rates from all training processes are scheduled to reduce by a factor of 10 at epochs 10 and 13. LeakyReLU and Layernorm are used as the activation function and normalization for each linear layer of every module. We also apply a training warm-up strategy on the first 5 epochs by increasing the learning rate by factor of 2 to the base learning rate at epoch 5. This helps reducing the impact of unstable physics simulation at the beginning of training, mitigating gradient explosion.

## 4.3 Metrics

There are two standard protocols for the evaluation on Human3.6M [15]. Both of these protocols assess the *Mean Per Joint Position Error* (MPJPE). This metric represents the average Euclidean distance between the reconstructed joint coordinates and the provided ground truth 3D keypoints. While the first protocol directly calculates the MPJPE for root-aligned poses, the second protocol initially employs a rigid alignment between the poses which is called MPJPE-PA (MPJPE Procrustes Aligned). Since our approach estimates poses in a global coordinate system, we additionally calculate the MPJPE-G in global coordinates which is the MPJPE without frame-wise root alignment. In addition to the different variations of the MPJPE, the *Percentage of Correct Keypoints* (PCK) measures the percentage of predicted joints that are within a distance of $150mm$ or less from their corresponding ground truth joint. Unlike the PCK, the CPS measurement [43] determines a pose as correct only if all its joints are estimated correctly according to a threshold value, similar to the PCK. To ensure independence from a specific threshold value, the CPS computes the area under the curve within the 1mm to 300mm threshold range. To evaluate the global translation error, not accounting for the differences between poses, we report the global root position (GRP) error, which calculates the Euclidean distance between only the root joints. We also use the acceleration (Accel) metric from [19] to measure the jitter of the output motions. Accel is computed as the second-order difference between 3D keypoints across all sequence frames.

## 4.4 Comparison with State of the Art

We report the quantitative results of *OSDCap* and other related work on different metrics in Tab. 1. Due to the novelty of dynamics-based motion capture the evaluation protocols differ significantly across different approaches. Here, we make an effort of consistently structuring approaches with similar evaluation protocols to achieve a fair comparison. To be as comparable as possible we follow the most used protocol introduced by Shimada et al. [37]. We outperform all online approaches in MPJPE, PCK and CPS. For the global error MPJPE-G, we improve upon state of the art by a large margin. Notably, DnD [21] achieves a lower MPJPE-PA. However, DnD's estimation depends on encoding the full action sequence, extracted from temporal convolutions, assuming significantly more knowledge which is not suitable for an online setting. Moreover, AMASS [25] is used as an additional training data source, thereby, not following the standard protocols for Human3.6M. SimPoE [50] achieves best smoothness performance on the Accel metric, due to being constrained by a high-frequency physics engine. However, as the discussion in Sec. 1, only relying on modeling the physics can lead to sub-optimal human pose quality. IPMAN-R [41] also shows good performance in terms of MPJPE-PA. However, it is a single-image approach that contains physically inspired constraints such as ground penetration, but no dynamics. The MPJPE-PA, i.e. the MPJPE after

| Methods | Phys. | Onl. | MPJPE ↓ [mm] | MPJPE-G ↓ [mm] | MPJPE-PA ↓ [mm] | PCK ↑ [%] | CPS ↑ [mm] | GRP ↓ [mm] | Accel ↓ [$mm/s^2$] |
|---|---|---|---|---|---|---|---|---|---|
| Vnect [27] | ✘ | ✔ | 89.6 | - | 62.7 | 85.1 | - | 185.1 | - |
| HMMR [17] | ✘ | ✔ | 79.4 | - | 55.0 | 88.4 | - | 231.1 | - |
| HMR [16] | ✘ | ✔ | 78.9 | - | 54.3 | 88.2 | - | 204.2 | - |
| TRACE [40] | ✘ | ✔ | 78.1 | 152.7 | 62.5 | 88.3 | 169.1 | 125.9 | 19.2 |
| VIBE [19] | ✘ | ✔ | 68.6 | 207.7 | 43.6 | - | - | - | 23.4 |
| Gärtner et al. [9] | ✔ | ✘ | 84.0 | 143.0 | 56.0 | - | - | - | - |
| DiffPhy [8] | ✔ | ✘ | 81.7 | 139.1 | 55.6 | - | - | - | - |
| PhysPT [52] | ✔ | ✘ | 52.7 | - | 36.7 | - | - | - | - |
| *DnD [21] | ✔ | ✘ | **52.5** | - | **35.5** | - | - | - | - |
| PhysCap [37] | ✔ | ✔ | 97.4 | - | 65.1 | 82.3 | - | 182.6 | - |
| NeurPhys [38] | ✔ | ✔ | 76.5 | - | 58.2 | 89.5 | - | - | - |
| Xie et al. [46] | ✔ | ✔ | 68.1 | - | - | - | - | **85.1** | - |
| IPMAN-R [41] | ✔ | ✔ | 60.7 | - | 41.1 | - | - | - | - |
| SimPoE [50] | ✔ | ✔ | 56.7 | - | 41.6 | - | - | - | **6.7** |
| OSDCap | ✔ | ✔ | 54.8±0.1 | **132.8**±1.6 | 39.8±0.1 | **95.5**±0.1 | **197.7**±0.1 | 119.1±1.8 | 8.4±0.2 |

Table 1: Quantitative comparison on the Human3.6M dataset [15]. Related methods are separated into two main categories: kinematics (top) and physics-based (bottom). In addition only [37, 38, 46, 50] retains the online prediction ability of the video-based kinematics estimations. Bold numbers denote the best evaluation score on each metric. Our approach achieves state-of-the-art in MPJPE and PCK among online approaches, and competitive results on GRP and Accel. Note that *DnD [21] does not follow standard evaluation protocols by using additional training data.

| Dataset | Methods | MPJPE ↓ [mm] | MPJPE-G ↓ [mm] | MPJPE-PA ↓ [mm] | PCK ↑ [%] | CPS ↑ [mm] | GRP ↓ [mm] | Accel ↓ [$mm/s^2$] |
|---|---|---|---|---|---|---|---|---|
| Fit3D [7] | TRACE [40] | 85.4 | 131.2 | 65.2 | 85.5 | 166.6 | 178.1 | 20.2 |
|  | OSDCap | **58.7** | **73.8** | **42.6** | **96.7** | **209.4** | **47.2** | **8.2** |
| SportsPose [14] | TRACE [40] | 97.3 | 361.9 | 71.1 | 60.1 | 168.1 | 333.0 | 15.8 |
|  | OSDCap | **71.7** | **113.6** | **52.4** | **68.8** | **190.0** | **90.2** | **10.9** |

Table 2: Evaluation results on Fit3D [7] and SportsPose [14]. *OSDCap* improves the kinematics baseline TRACE by a large margin across all metrics. We fine-tune *OSDCap* (pretrained on Human3.6M) on SportsPose's ground truth keypoints for additional 15 epochs. Even with very noisy inputs from SportsPose, *OSDCap* still manage to retain the robust estimation thanks to the Kalman filtering process, especially on global translation metrics (MPJPE-G and GRP).

pose-wise rigid alignment, is reported for completeness. While being a reasonable metric for single-image pose estimation, we argue that for physics-based pose estimation, rigid alignments distort the interpretation of the results since they remove all information about global rotation.

We additionally evaluate *OSDCap* on the more challenging motions in the Fit3D dataset [7]. Tab. 2 shows the results. Since Fit3D is recorded in the same setting as Human3.6M, we additionally evaluate on the newer SportsPose [14] dataset to show the generalizability to other motion domains. We improve on the kinematics baseline TRACE by a large margin, especially for global metrics as shown by the MPJPE-G and GRP. Fig. 3 illustrates the benefits of *OSDCap*. *OSDCap* significantly reduces the impact of noisy and inaccurate kinematics input when encountering high depth-uncertainty from monocular views, while retaining the correct estimation with respect to the ground truth. The bars on the left represent the predicted Kalman gains, where the y-direction is the direction of the optical axis which indicates a predicted low trust in the kinematics prediction and leads the Kalman filter to prefer the physics simulation. Fig. 3b shows an example (side view) of *OSDCap* adjusting unnatural leaning into a physically plausible pose by our physics simulation.

## 4.5 Ablation study

### 4.5.1 Optimal-State Estimation

We conduct an ablation study to verify the impact of the optimal-state estimation process on simulated motions. We sample a subset of data consisting of only the first action class of all subjects in the camera view 60457274 from Human3.6M [15]. S9 and S11 are for evaluation, and the rest for training. This setup creates a suitable challenge to test the proposed method, limiting the types of motion that are seen during training. Tab. 3 shows the original simulated result from straight-foward smoothing methods, PD controller and the improvement by *OSDCap*.

| Methods | #params. [$mm$] | MPJPE ↓ [$mm$] | MPJPE-G ↓ [$mm$] | MPJPE-PA ↓ [%] | PCK ↑ [$mm$] | GRP ↓ [$mm/s^2$] | Accel ↓ |
|---|---|---|---|---|---|---|---|
| TRACE [40] | - | 78.4 | 153.9 | 62.7 | 88.1 | 128.2 | 19.7 |
| TRACE (median) | - | 78.2 | 153.1 | 62.6 | 88.2 | 127.4 | 13.6 |
| TRACE (Gaussian) | - | 77.8 | 162.4 | 62.4 | 88.5 | 126.7 | 6.5 |
| PD (only) | 8.4M | 87.7 | 145.0 | 67.7 | 82.7 | 105.9 | 6.4 |
| PD (Gaussian) | 8.4M | 77.7 | 136.0 | 61.0 | 86.5 | 103.2 | **5.2** |
| OSDCap (no bias) | 6.6M | 55.0 | 111.9 | 40.0 | 95.7 | 94.9 | 9.5 |
| OSDCap | 7.2M | **54.0** | **111.0** | **40.0** | **95.9** | **94.8** | 8.7 |

Table 3: Ablation study on the impact of *OSDNet* on a subset of Human 3.6M [15]. Naive methods such as median or Gaussian smoothing cannot help with the plausibility of the pose. Without our Kalman filtering process, the PD controller cannot train and estimate the correct dynamics. We also study the effects of the inertia-bias $M^b$ and some performance gains has been recorded.

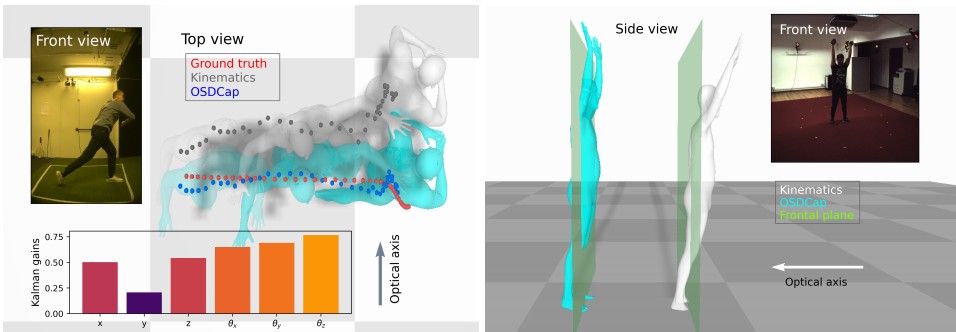

(a) Incorrect global translation.   (b) Unnatural leaning artifacts.

Figure 3: Qualitative results of *OSDCap* (cyan) compared to the kinematics input [40] (purple), with corresponding ground truth pose (red). Left: Filtering results of *OSDCap* on a sample from SportsPose [14], where the kinematics estimation is very inaccurate along the camera's depth dimension. The Kalman gain at the y-axis (optical axis) is greatly decreased due to the incorrect translation of the kinematics input. Therefore, the simulated state is preferred. Right: Example from Fit3D [7], with an unnaturally leaning pose caused by depth ambiguities. Unlike Fig. 3a, the three poses are manually separated apart for better visualization. *OSDCap* recovers the physically plausible upright pose.

Naive approaches for smoothing the noisy input estimation apply temporal filters such as median or Gaussian filter. However, simply filtering the signal does not help the motion to become physically plausible, unnatural poses still prevail. As shown in Tab. 3, naive filtering reduces the jitter of the input motions (reduction in Accel measurements), but does not help with any other metrics.

To ensure the plausible physics constraints of the forward dynamics process (unlike [37, 38]), we employ external forces into the calculation, which leads to a much more challenging scenario for the PD controller. This can be observed in Tab. 3 where the PD controller struggles to reconstruct the motions, even with temporal filtering on the input signals and increase the number of parameters. By using our optimal-state estimation module, the PD controller has a significantly better performance, leading to the optimal results for online human motion reconstruction.

### 4.5.2   Comparison to classical Kalman Filter

The biggest challenge of using classical Kalman filter for *OSDCap* is the tuning of unknown noise covariances of both the kinematic input TRACE [40] and the simulated result from PD controller. Our choice of a learnable Kalman filter [36] relieves us from trial-and-error process of finding the correct noise covariance matrices and achieves the best results. We conducted an additional experiment where we replace our learnable filter by a traditional one, the results are shown in Tab. 4.

Assuming noise covariances that are constant over time and equal in all directions, the ratio between the noise covariance of the simulated PD controller (process noise) and the noise covariance of the kinematic input TRACE (measurement noise) governs the quality of the Kalman filter estimates. The evaluation results can be seen in Tab. 4, where we use constant noise covariances with ratios $100/1, 10/1, 1/1, 1/10, 1/100$ between process noise and measurement noise. While a classical Kalman filter approach increases the result marginally, optimal results are difficult to find.

| Method | MPJPE ↓ [mm] | MPJPE-G ↓ [mm] | MPJPE-PA ↓ [mm] | PCK ↑ [%] | GRP ↓ [mm] | Accel ↓ $[mm/s^2]$ |
|---|---|---|---|---|---|---|
| TRACE | 78.4 | 153.9 | 62.7 | 88.1 | 128.2 | 19.7 |
| cKF_kin_only | 78.3 | 153.0 | 63.0 | 87.9 | 127.4 | 7.8 |
| cKF_100/1 | 60.9 | 120.7 | 43.4 | 94.3 | 102.0 | 7.7 |
| cKF_10/1 | 61.5 | 122.6 | 43.7 | 94.4 | 103.0 | 9.1 |
| cKF_1/1 | 59.9 | 117.6 | 43.2 | 94.8 | 100.1 | **6.5** |
| cKF_1/10 | 63.6 | 124.0 | 44.3 | 93.8 | 102.7 | 11.5 |
| cKF_1/100 | 65.3 | 132.3 | 44.0 | 93.5 | 110.2 | 9.7 |
| OSDCap | **54.0** | **111.0** | **40.0** | **95.9** | **94.8** | 8.7 |

Table 4: Ablation study on the performance of the classical Kalman filtering (cKF) on the ablation set from the Human 3.6m dataset. Due to unknown noise covariance matrices, we tested with constant noise covariances with ratios $100/1, 10/1, 1/1, 1/10, 1/100$. The performance of applying Kalman filtering on only the kinematics input TRACE [40] (cKF_kin_only) is also conducted.

| Method | GP ↓ [mm] | GD ↓ [mm] | Friction ↓ [mm] | Velocity ↓ $[mm/s]$ | Foot-skating ↓ [%] |
|---|---|---|---|---|---|
| TRACE | **2.6** | 12.5 | 31.5 | 22.4 | 37.0 |
| OSDCap | 5.3 | **8.2** | **14.6** | **12.8** | **15.2** |

Table 5: Additional physics-based measurements for kinematics input TRACE and *OSDCap*. Because the ground penetration (GP) metric does not correctly reflect the foot-ground contact quality, i.e. floating above the ground is ignored and produces no error, we propose using an additional ground-distance (GD) metric. For foot-skating, we followed DiffPhy to compute the percentage of frames that contain skating artifacts over the whole sequence.

### 4.5.3 Additional physics-based metrics

We provide additional metrics for physic-based measurements introduced in Sec. 3.3. The results can be seen in Tab. 5. *OSDCap* helps refining the input kinematics on most of the physics-based metrics. Note that TRACE[40] outperforms our approach in the ground penetration metric. The reason is that in most cases the TRACE predictions float above the ground, which gives a low penetration error but can be seen as equally bad. Thus, we additionally provide a ground distance metric (GD) to reflect the correct foot-ground quality during contact. The value is computed as the mean absolute vertical differences between foot contact points and ground plane during contact duration, expressed as

$$\frac{1}{6} \sum_{c=1}^{6} \rho_c |p_c^{OSD} - p_c^{GT}|, \tag{12}$$

where $\rho_c$ is the predicted binary label of contact, $p_c^{OSD}$ and $p_c^{GT}$ are the 3D vertical positions of contact. There are a total of six contact points considered, three contacts in each foot accounting for heel, foot and toe. The joint configuration follow Human 3.6M skeleton [15], with bone length between joints optimized during training and fixed during inference.

## 5 Conclusion

This paper presents *OSDCap*, a new physics-based approach to reconstruct kinematics-based human motion captured from monocular videos. We found that previous approaches relying only on a physical simulation produce non-optimal motions due to unavoidable imperfections in the physical model and noisy measurements. This led us to introduce a learnable Kalman filtering for refining implausible motions simulated by a PD controller with noisy kinematic evidence as the target. In comparison with related research on physics-based motion capture, the proposed approach achieves state-of-the-art results on the Human3.6M, Fit3D, and SportPose datasets, especially on the global estimation of pose trajectories.

**Limitations and future work.** While taking a step into highly accurate predictions of the full body dynamics, our physical external forces are still not comparable to directly measuring with mechanical force plates. However, our approach only requires a single camera, e.g. from a smartphone, instead of a motion capture studio or other expensive hardware, such as force plates, to estimate meaningful forces. In the future, detailed modeling for the hands, feet, and body shape, will be investigated, targeting more realistic motion reconstruction.

## Acknowledgments and Disclosure of Funding

This research is partially supported by the Wallenberg Artificial Intelligence, Autonomous Systems and Software Program (WASP), funded by Knut and Alice Wallenberg Foundation, and by the Sensor AI Lab, under the AI Labs program of Delft University of Technology. The computational resources were provided by the National Academic Infrastructure for Supercomputing in Sweden (NAISS) at C3SE, and by the Berzelius resource, provided by the Knut and Alice Wallenberg Foundation at the National Supercomputer Centre.

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

# Supplementary Materials

## A    Network details

Fig. 4 shows the architecture of the proposed *OSDNet*. The network consists of multiple branches for Kalman gains, PD controller gains, inertia-bias, and external force estimations. The Kalman estimation module takes the combined inputs from the GRU's and current states embedding, outputting the Kalman gains matrix. The diagonal of Kalman gain matrix is initialized to be approximately $0.5$ by modifying the bias of the last linear layer. This is due to the instability of the PD branch at the beginning of training, which may cause gradient explosion if the Kalman gains are too low (zero trust in the kinematics stream). The hidden states $h_{\mathrm{GRU}}$ of the GRU unit are updated throughout the simulated sequence and used as one of the inputs for the next prediction.

Similar to [38], we scale $\boldsymbol{\kappa}_P$ and $\boldsymbol{\kappa}_D$ differently for global translation, global rotation, and joint angles. The initial scaling are $[30.0, 14.0, 1.9]$ for $\boldsymbol{\kappa}_P$ and $[1.5, 0.1, 0.05]$ for $\boldsymbol{\kappa}_D$. Notice that our initial gains are much lower than [38], because we want to explain the global motion by external reaction forces, avoiding the need for "unrealistic" residual force. These scalings are further optimized along with the training of *OSDNet*.

*OSDNet* estimates the inertia-bias matrix $\mathbf{M}_t^b$. To ensure the symmetric positive definite (SPD) of the inertia matrix, we estimate an intermediate $\mathbf{M}_{\mathrm{base}\,t}^b$ and compute $\mathbf{M}_t^b = \mathbf{M}_{\mathrm{base}\,t}^b + (\mathbf{M}_{\mathrm{base}\,t}^b)^\intercal$.

The Jacobian branch of *OSDNet* takes the current state embedding as input and outputs the Jacobian matrix that maps end-effector linear velocity to rotational velocity of each joints. The contact and external force branch takes the current feet positions and velocity as additional inputs to the state embedding. The contact branch outputs are mapped by a sigmoid function to create the contact probability $\rho_t^c$. The external force branch outputs the linear reaction force for two feet, with the vertical-axis initialized with the weight $(9.81 * \mathrm{mass})$ of the proxy character.

The adaptation matrix $\mathbf{C}$ and observation matrix $\mathbf{H}$ are initialized as identity matrices. We optimize them during the training process, and they are kept constant during inference.

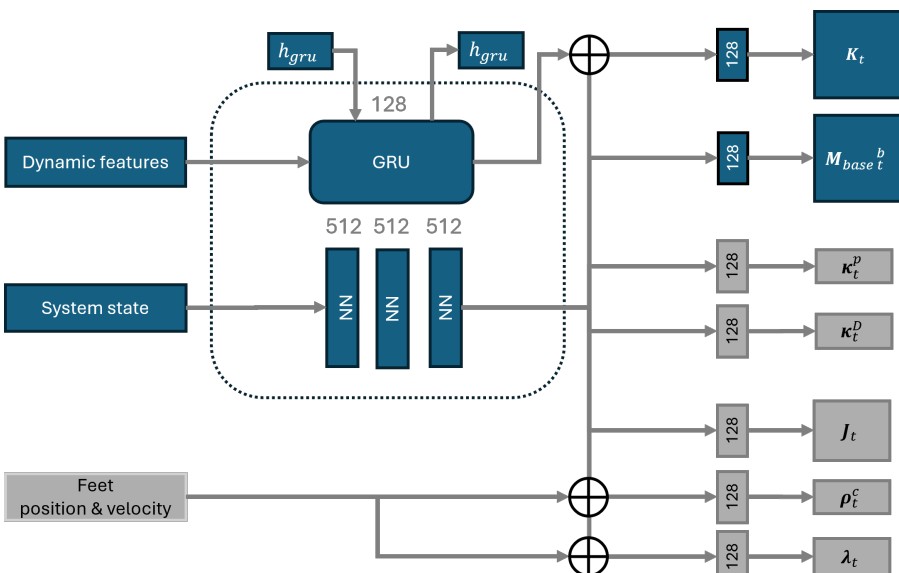

Figure 4: Architecture of the proposed *OSDNet*. The network consists of 3 hidden layer of size 512 to generate system state's embedding. Based on the state embedding, the inertia-bias matrix $\mathbf{M}_{\mathrm{base}\,t}^b$, PD gains $\boldsymbol{\kappa}_P, \boldsymbol{\kappa}_D$, Jacobian matrix $\mathbf{J}_t$, contact probability $\rho_t^c$ and external force $\boldsymbol{\lambda}_t$ are estimated. The proposed GRU unit with size 128 takes the dynamics features (mentioned in Sec. 3.2) as input, the Kalman gain matrix $\mathbf{K}_t$ is estimated from the concatenation of GRU and the state embedding. The hidden state $h_{\mathrm{gru}}$ is continuously updated at each time step. For a better estimation of foot-ground contacts and reaction forces, we also feed the feet position and linear velocity as additional inputs.

## B  Human body proxy model

The simulated proxy character is created based on the body configuration of the SMPL model [24]. Bone lengths and weight distribution are the same as in the Human 3.6M metadata of the 'common' human body [15]. Fig. 5 is a visualization of the proxy character, composed of spheres and cylinders. The bone lengths are treated as extra learnable parameters and optimized along with the model during the training process. During testing, the bone lengths are fixed to the ones learned during training, i.e. no ground truth bone lengths are used when testing.

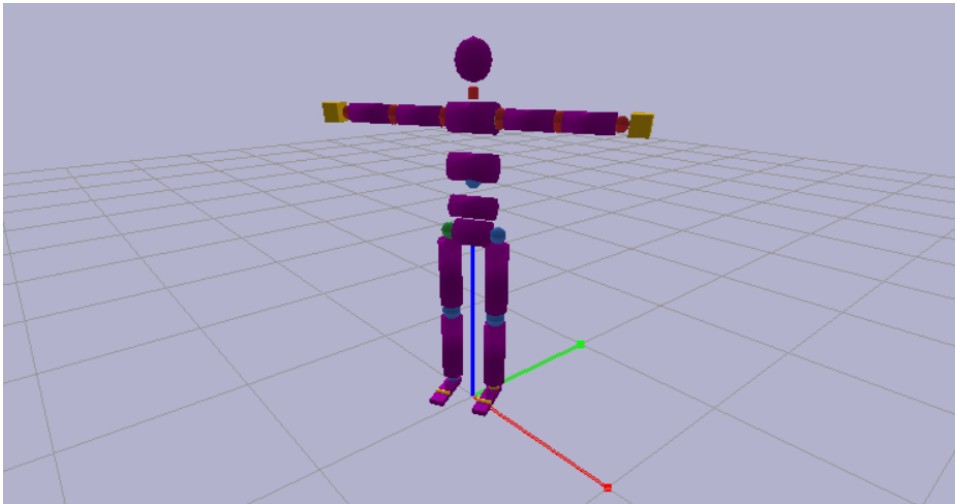

Figure 5: The simulated proxy character used in the paper. The RBDL library [6] is used to extract the inertia matrix $M_t$ and bias forces $h(q, \dot{q})$.

## C  Dataset details

As mention in Sec. 4.1, we evaluate our proposed method on Human3.6M [15], Fit3D [7], and SportsPose [14]. For Human3.6M, we follow prior works [37, 38, 21] to consider actions that only involve foot-ground contact (S1, S5, S6, S7, S8) for training and (S9, S11) for testing. For [7], we apply the same protocol with only foot-ground available actions are used: (s03, s04, s05, s07, s08, s10) for training and (s09, s11) for testing. For SportsPose, we only consider sequences that contain human at time step 0: (S02, S03, S05, S06, S07, S08, S09) for fine-tuning and (S12, S13, S14) for evaluation.

TRACE [40] is used to extract the kinematics input from the data. All extracted motions are aligned at the origin in the first time step, eliminating the effect of wrongly calibration process. Each action is then equally split into 100-frame sub-sequences, utilizing batch processing for training the *OSDNet*.

## D  Contact labels

Since there are ground truth contact labels are provided in all three datasets [15, 7, 14], we generate our own annotations based on the ground truth keypoints. To create contact labels, ground truth feet 3D positions are considered. If a foot position is within 10 cm (already compensated for shoes and inconsistent MoCap sensor placement) above the ground plane and also not moved more than 2cm from the previous frame, it is labeled as a valid contact point. Similar protocol is applied for both left and right foot. The foot-ground contacts are modelled directly by the OSDNet and automatic data annotation, using the ground truth 3D poses from the training data set.

## E  Global rotation

To mitigate the Gimbal lock problem in the original Euler representation of Human3.6M [15], we convert all root rotation (d.o.f $3^{\text{rd}}$ to $6^{\text{th}}$ of the state vector $q_{t|t}$) into quaternions $quat_{t|t} = (x, y, z, w)$,

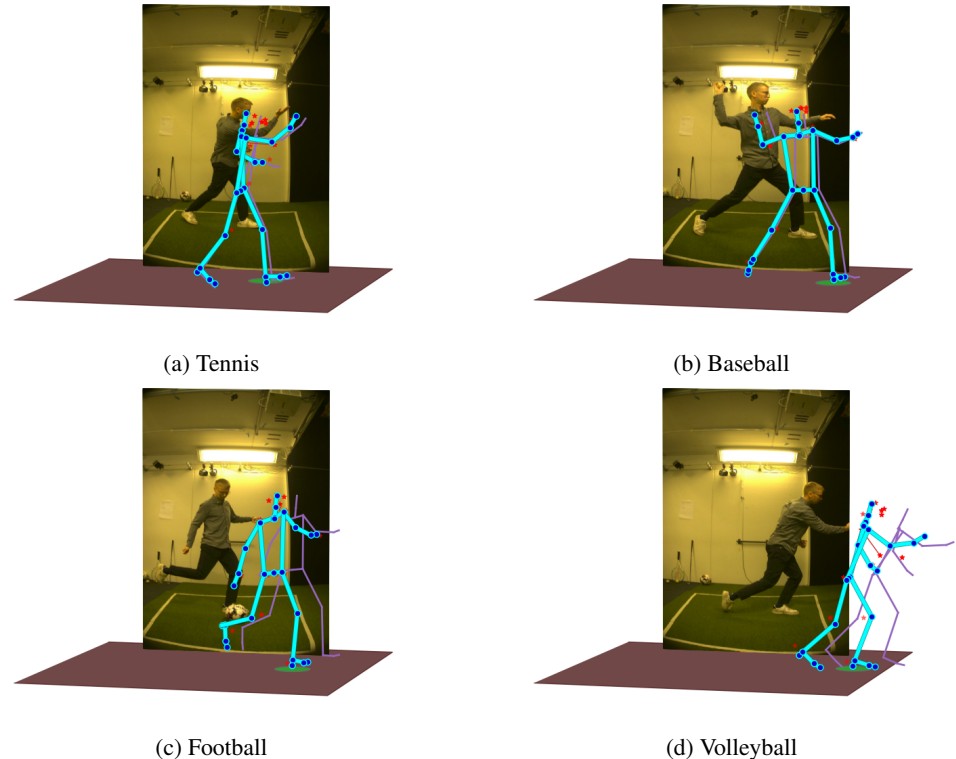

(a) Tennis        (b) Baseball

(c) Football        (d) Volleyball

Figure 6: Example results on SportsPose [14] test data. Here we show four out of five action classes of SportsPose [14] that have foot-ground contacts. Qualitatively *OSDCap* matches the provided ground truth much better than the kinematics input TRACE.

with the real part at the end. Since quaternions are not a linear representation, the computation of quaternion differences is given as

$$\Delta quat_{t|t} = quat_{t+1|t+1} * quat_{t|t}^{-1}. \tag{13}$$

$\Delta quat_{t|t}$ is the input error term for the PD controller for computing the corresponding root torque by Eq. 5 and Eq. 1. The procedure for finite integration (during the state transitioning stage) from state vector $\mathbf{q}_{t|t}^{quat}$ to the predict state $\mathbf{q}_{t+1|t}^{quat}$ given the system state vector $\dot{\mathbf{q}}_t^{quat}$ in quaternions is expressed as

$$\mathbf{q}_{t+1|t}^{quat} = \mathbf{q}_{t|t}^{quat} + 0.5(\dot{\mathbf{q}}_t^{quat} * \mathbf{q}_{t|t}^{quat})\Delta t. \tag{14}$$

## F    Computing resources

The proposed pipeline of *OSDCap* was trained and evaluated on the NVIDIA-A100 GPU with 40Gb of memory. In average, *OSDCap* requires an additional 0.02 second on top of the processing time of the kinematics estimation [40] on each frame. Each ablation study in 4.5 takes 45 minutes to train and evaluate. The full training and testing on Human3.6M consumes approximately 2 hours, on Fit3D 1 hour, and on SportsPose 15 minutes.

Besides, we also train and evaluate *OSDCap* on multiple random seed values to demonstrate the reproducibility of results. Table 1 presents the means and standard deviations of the evaluation results across multiple random seeds from 0 to 4. We did not report error bars for every other experiment since it would be too computationally expensive.

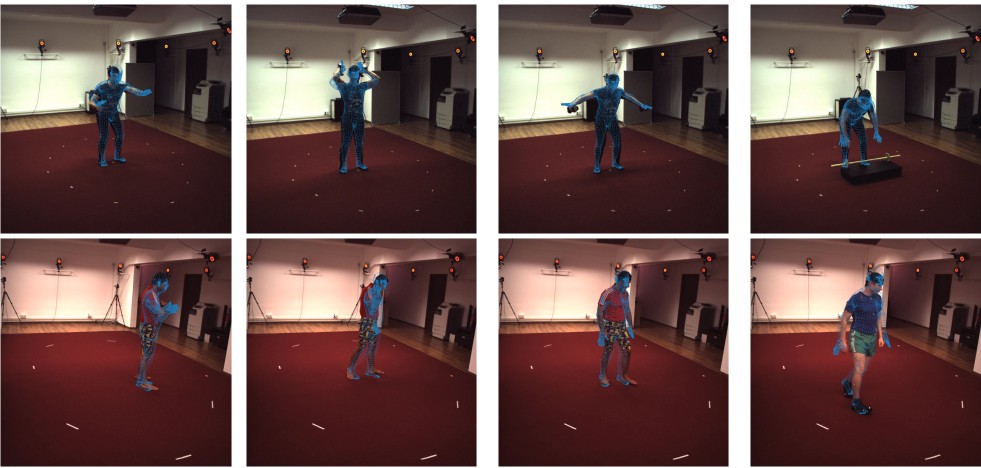

Figure 7: Qualitative results when projecting the SMPL body model from the *OSDCap* poses back to the input 2D images. The overlayed SMPL models are shown in sparse blue point cloud to maximize the visibility of the input human pose.

## G    Additional results

One can refer to our additional supplementary material for a better visualization of the *OSDCap* reconstructions against the kinematics input and ground truth. Some example footage on the challenging SportsPose dataset can be seen in Fig. 6.

Additional visualization of estimated pose overlayed on 2D input images can be found in Fig. 7. There is always a trade-off between the reprojection error and the model-based assumptions. In our case the physics simulation uses stronger assumptions than a purely kinematics-based model. On the other hand, it produces more plausible motion as shown in Tab. 1 of the main paper and Tab. 5.

Despite not being a training objective during training, the re-projected poses match well with the input humans in the input image as shown in Fig. 7. The slight offset is due to the mis-match bone length between the proxy character and the actual testing human subjects. An adaptive human shape estimation would be investigated in the future.

