# OpenReview forum: "Optimal-state Dynamics Estimation for Physics-based Human Motion Capture from Videos"
_NeurIPS.cc/2024/Conference — NeurIPS 2024 poster_

### Official Review · Reviewer_mMU1 · 2024-07-09

**Soundness:** 3
**Presentation:** 3
**Contribution:** 2
**Rating:** 5
**Confidence:** 5

**Summary:**

This paper proposes a method to capture physically plausible human motions from monocular videos. The paper builds on top of NeuralPhys and introduces Kalman filtering to alleviate noise from kinematic motions. The framework combines Kalman filtering and physics simulation to be fully differentiable, thus enabling training in a supervised manner. The experimental results show that the proposed approach can achieve competitive performance.

**Strengths:**

- The proposed method uses Kalman filtering to produce targets for PD control, which is more robust to the noise from pure kinematic poses.
- The framework can be trained in a supervised manner due to its fully differentiable implementation.

**Weaknesses:**

- The simulation and filtering refine the initial kinematic poses with dynamic constraints, which do not consider the image observations. The simulation may remove some high-frequency motions; thus, the refined poses may not be consistent with 2D poses. The authors should overlay the 3D pose onto 2D images and show more visualization results so that we can evaluate the discrepancy.
- Although the method predicts contact information, the framework cannot prevent penetrations between humans and the floor since scene dynamics are not considered.
- The optimal pose is predicted from Kalman filtering, while the velocity is obtained from simulation. In previous works, the final body pose is integrated from simulated velocity as in Eq. 2, which can guarantee physical plausibility. However, the optimal pose in this work is further updated with Kalman filtering, which may not be consistent with simulated velocity. Therefore, the final poses may not obey physical laws.
- The work relies on external force, which may result in some unnatural states (e.g., body leaning).

**Questions:**

- Do $q_t$, $\dot{q}_t$ in the output still follow rigid body dynamics as in Eq. 1?
- The work builds on top of NeuralPhys with additional Kalman filtering. What is the performance of applying Kalman filtering on simulated or kinematic motion only?
- Humans may penetrate the ground plane due to depth ambiguity. How does the method address penetration between humans and the ground plane?
- The body shape in this work is fixed. Can the trained models be generalized to different skeleton shapes?

**Limitations:**

The limitations are discussed.

---

> ### Author Rebuttal · Authors · 2024-08-06
>
> We thank the reviewer for the thoughtful and constructive feedback. We address the concerns and questions from the reviewer in the section below.
>
> **2D overlay of estimated pose on the input image.**
>
> Thank you for the suggestion. We plotted the overlay of 3D estimated poses from OSDCap onto the input 2D images in Figure*1. As can be seen from the figure, the re-projected SMPL meshes overlay closely to the human inside the input images. Despite the image-based observations being refined by the Kalman filter, the idea of considering original image observations could potentially benefit the prediction, and will be investigated in future work.
>
> **OSDCap cannot prevent foot-ground penetration and scene dynamics.**
>
> Yes, foot-ground contact is not explicitly modeled in OSDCap. However, our contact estimation is able to mitigate it and creates physically plausible motion. Please refer to Table*2 in the attached PDF above for our additional physics-based measurements of OSDCap.
>
> **Is the final pose obeying physics laws?**
>
> OSDCap combines the two data sources and collectively creates the best of both worlds. Naturally, if the GRU model decides that the noise from the physics simulation data is larger than the kinematic input in a specific time step, the resulting human pose $q_t$ and $\dot{q}_t$ after Kalman filtering might not strictly obey physics laws anymore. There always exists a knowledge gap between simulation and real-world that can cause the simulated result to be sub-optimal under the provided constraints. While the image-based kinematic estimation, despite being jittery and noise, reflects real-world observation and can act as an online compensation for the mistakes made by the simulation.
>
> **Unnatural leaning due to the usage of external force.**
>
> A false external force prediction might result in implausible artifacts such as unnatural leaning. This is one of the motivation for proposing a Kalman filtering approach, where a noisy inputs to the in physics simulation, i.e. false external force prediction, can be monitored and compensated directly by the kinematics input. However, unnatural leaning artifacts often occur more in the kinematic input due to depth uncertainty in the optical axis, especially in scenarios when the person is facing the camera. We show an example of using a physics-based simulation with external forces can actually refine the unnatural pose and prevent unnatural leaning in Figure 3b in the main paper.
>
> **Kalman filter on kinematics or simulation only.**
>
> The performance of applying Kalman filtering on kinematic motion can only be shown in Table*1 in the rebuttal PDF above. As can be seen, without physics-based knowledge, the Kalman filtering scheme only does smoothing and achieves worse performance than our approach. For a deeper analysis of the classical Kalman filter compare to a learnable one proposed in the paper, we kindly refer to the discussion with reviewer wqMZ.
>
> **Adaptability of trained models on different skeleton shapes.**
>
> The trained models can be generalized to different skeleton shapes since we are predicting joint angles of the skeletons while training the NN models. Therefore, a motion retargeting to a character with different bone lengths is straight-forward. However, achieving the same performance as in the three used dataset Human3.6M, Fit3D and SportPose for data with a completely different character configuration would require additional training.

---

> > ### Comment · Reviewer_mMU1 · 2024-08-13
> >
> > Thanks for the rebuttal, which has addressed most of my concerns. However, the rendered images show that the model-image alignment is inferior to purely kinematics-based methods. I hope the authors can discuss the impact of dynamics and filtering on model-image alignment and joint accuracy.
> >
> > The proposed method cannot guarantee 100% physical plausibility due to the lack of contact modeling and the use of additional filtering. These limitations should be included in the paper.
> >
> > Additionally, there are existing works [A] that adopt filters to improve dynamics-based methods. The differences between these approaches should be discussed.
> >
> > [A] Xie, Kaixiang, and Paul G. Kry. "Inverse Dynamics Filtering for Sampling‐based Motion Control." Computer Graphics Forum. Vol. 40. No. 6. 2021.

---

> > > ### Author Response · Authors · 2024-08-13
> > >
> > > We thank the reviewer for the feedback.
> > >
> > > **Model-image alignment**
> > >
> > > There is always a trade-off between the reprojection error and the model-based assumptions. In our case the physics simulation uses stronger assumptions than a purely kinematics-based model. On the other hand, it produces more plausible motion as shown in Table 1 of the main paper and Table*2 in the rebuttal pdf above.
> > >
> > > Despite not being a training objective during training, the re-projected poses match well with the input humans in the input image as shown in Figure*1. The slight offset is due to the mis-match bone length between the proxy character and the actual testing human subjects. An adaptive human shape estimation would be investigated in the future.
> > >
> > > **Contact modeling**
> > >
> > > We modelled contact through automatic data annotations and predictions from a neural network. Sophisticated contact modelling (i.e. from physics engines) would limit the calculation of a gradient for an end-to-end approach such as ours. However, our simple contact modeling still demonstrates the ability to increase the physical plausibility of the input kinematics as can be observed in Table 1 and Table*1. We will add a discussion to the main paper.
> > >
> > > **Additional reference**
> > >
> > > Thank you for the additional reference. The Butterworth filtering method used in the paper [A] from Xie and Kry is a good idea for filtering out noisy kinematics inputs before computing inverse dynamics. However, the method requires an empirical selection of cutoff thresholds for each type of motion separately, thus limiting the scalability of the method to different data domains and environmental setups. In contrast, OSDCap provides an end-to-end approach to effectively integrate dynamics information for refining noisy kinematics in an online manner, which is very valuable for real-world applications.
> > >
> > > We would like to highlight again that our proposed approach introduces a novel integration of learnable Kalman filters, thereby mitigating problems of previous approaches in physics-based motion capture and producing state-of-the-art results. While we agree with the reviewer’s contact modelling comments, we like to encourage a weighting of the novelty against the non-perfect contact model.

---

> > > > ### Comment · Reviewer_mMU1 · 2024-08-14
> > > >
> > > > Thanks for the clarifications.

---

> > > > > ### Author Response · Authors · 2024-08-14
> > > > >
> > > > > We thank the reviewer for the productive discussion and the upgrade of the rating.

---

### Official Review · Reviewer_QEqx · 2024-07-10

**Soundness:** 3
**Presentation:** 4
**Contribution:** 3
**Rating:** 7
**Confidence:** 4

**Summary:**

The paper presents a novel physics-based human motion capture method that is physically explainable, conforming to the PD control theory and rigid body dynamics. The key designs of the method involve an integration of a kinematic Kalman filter and Newtonian equation-based physics simulation, and learnable Kalman gains, PD gains, external forces and robot inertia biases. The method outperforms existing kinematics-based and physics-based motion capture methods on keypoint accuracies.

**Strengths:**

(1) The method is physically explainable without unrealistic approximations of the control process and the robot dynamics.

(2) Under the paradigm of using physics simulation to capture human motion, the method provides novel insights about which physical properties should be modeled by neural networks.

(3) The method is superior to previous kinematics-based and physics-based methods in the accuracy of joint predictions.

(4) The writing is clear and easy to follow.

**Weaknesses:**

(1) The contact modeling only considers foot-ground contact, ignoring full-body contact that commonly appears in human-object and human-scene interaction scenarios. Besides, the contact on each foot is represented as a force vector on a pre-defined contact point, ignoring changes in the contact point and the resultant torque of the contact.

(2) The method updates the inertia matrix $M$ online. However, the inertia matrix is the attribute of the robot and should be fixed values during the whole motion capture process for better physical interpretability.

(3) To fully examine the generalizability of the proposed method, existing physics-based methods should also be compared on datasets Fit3D and SportsPose.

**Questions:**

Typos:

* In Figure 2, "$q_{t|t-1}$" -> "$q_{t+1|t}$"
* In the caption of Figure 2, "performs" is redundant
* In Equation 6, "$\sum_c^2$" -> "$\sum_{c=1}^2$"
* In line 295, "HMDCap" -> "OSDCap"
* In line 478, "the" is redundant

**Limitations:**

One limitation is that the human dynamic model is formulated as a connection of circles and cylinders, which neglects the modeling of geometric details and wearings of humans.

---

> ### Author Rebuttal · Authors · 2024-08-06
>
> We thank the reviewer for the highly positive evaluation and the constructive feedback to our work. For a better clarification, we provide the answers to the reviewer's questions below.
>
> **Full-body contacts.**
>
> We agree that full body contact is the next logical step towards a fully environment-aware physical model. Note that our approach (though not integrated in the current version) can easily add an additional force to any part of the body. However, this would require contact detection and estimation of the physical properties (e.g. velocity, weight, softness) of the object which is non-trivial. Recently, this field received more attention [C, D] and we are looking forward to integrating these approaches into our model in the future.
>
> For a similar discussion on body models that enable to add external forces we kindly refer to the “Selection of a simple proxy character” section with reviewer BVYQ.
>
> [C] Tripathi et al., DECO: Dense Estimation of 3D Human-Scene Contact In The Wild, ICCV, 2023.
>
> [D] Xie et al., Template Free Reconstruction of Human-object Interaction with Procedural Interaction Generation, CVPR, 2024.
>
> **Inertia-matrix M.**
>
> For a robotic arm, the mass distribution often does not change significantly. However, while a constant limb-wise mass distribution is an acceptable approximation for a human, it is not entirely correct. Due to muscle movement and soft tissue deformation the mass distribution changes which has a direct effect on the estimated motion [E, F]. While our inertia matrix intentionally leaves this additional degree of freedom, we agree that its interpretability is limited. Increasing the interpretability of the inertia matrix will be investigated as the next step of this work.
>
> [E] Pai, D.K., 2010. Muscle mass in musculoskeletal models. Journal of biomechanics 43, 2093–2098.
>
> [F] Featherstone, 2008. Dynamics of rigid body systems. Rigid Body Dynamics Algorithms, 39–64.
>
> **Evaluation of related works on Fit3D and SportPose.**
>
> Due to the availability of the implementation code of related physics-based methods, we could not replicate all related physics-based methods on Fit3D and SportPose during the rebuttal phase. We hope that these experiments will be finished during the discussion phase and at the latest we will add the evaluation results of works that provided implementation, such as NeurPhys, to the camera-ready version of the paper.
>
> **Typos.**
>
> Thanks for catching the typos. We will correct them.

---

> > ### Comment · Reviewer_QEqx · 2024-08-13
> >
> > Thanks for the authors' detailed rebuttal. I understand that exploring full-body contact and increasing the interpretability of the inertia matrix could be future work due to their challenges, and I am looking forward to the evaluations on existing physics-based methods. Nevertheless, I believe this work is theoretically valuable and useful to the research field of human motion capture.

---

> > > ### Author Response · Authors · 2024-08-13
> > >
> > > We thank the reviewer for the feedback and positive assessment.
> > >
> > > We are expecting the evaluation of the related works to be ready soon.

---

### Official Review · Reviewer_BVYQ · 2024-07-11

**Soundness:** 3
**Presentation:** 3
**Contribution:** 4
**Rating:** 8
**Confidence:** 5

**Summary:**

This work focuses on tackling the problem of single-person motion estimation from a monocular video. Current approaches produce temporal artifacts such as jittering. Most approaches are entirely kinematic while others that combine physics, do it by re-simulating the kinematic inputs by using automatic PD controllers. These methods, however, require simplifying assumptions that compromise the motion realism.This method combines information from two sources to produce an optimal estimate of single-person 3D human motion. One source comes from an off-the-shelf kinematic motion estimation pipeline and the other from a differentiable physics formulation. These two "measurements" about the motion are combined in a Kalman-filter to generate an optimal output.  Here, the authors propose to selectively incorporate the physics models with the kinematics observations in an online setting, taking as inspiration the neural Kalman-filter. The method uses a meta-PD controller and a physics-based simulation step. The Kalman filter is realized via a recurrent neural network which aims to balance the kinematic inputs with the simulation.

The authors propose an end-to-end model for this purpose which is not trivial to accomplish. The method is capable of capturing accurate global trajectories and, at the same time, producing physically plausible human poses.

**Strengths:**

* The paper is very well written and the experiments are well presented which makes the paper easy to follow.
* Authors present extensive experiments comparing several SoTA methods and include in-domain and out-of-domain test data for these.
* The setup and the method are sound.
* I would say that this is the first paper that successfully combines information from kinematic estimates and a differentiable physics simulation step in an end-to-end manner. It is not trivial to refine kinematic estimates with physics simulation (or physics informed estimates) outside the RL framework. It seems that the neural Kalmal filter is a promising direction to bridge the gap between kinematics and physics. I believe that this work is of high significance for the field.

**Weaknesses:**

### **Presentation**
The qualitative results could be better presented as it is sometimes hard to have a good sense of the pose estimated by the kinematic approach (TRACE) both in Fig. 6 and in the supplementary .gif images. The way the poses and the original video are visualized can be improved. First, I would suggest making the kinematic skeleton more visible as it is “obscured” by OSDCap results. Even better, it would be nice to have SMPL visualizations of GT, kinematics and OSDCap as it is presented in Fig. 1. In my opinion, changing visualization styles within the paper can reduce the presentation quality. I also advise the authors to focus on video results with several examples. If possible I would like to see this as part of the rebuttal, if not, then this should be present for the camera ready version of the paper.

### **Physics-based metrics**
Sec 4.3: It would be interesting to show the results for more physics-based metrics other than the Acceleration metric, for example, foot skating and ground penetration, which should be corrected by the physics formulation. As authors are modeling contacts and have physics-based losses (e.g., friction, velocity), it would be interesting to know these metrics in comparison with the SoTA or at least the baseline used (TRACE).

### Minor
- (Typo) Fig.2: 4th line: performs contains-->contains.
- (Typo) L295: HMDCap -->OSDCap?

### References
I would advise the authors to include most recent papers that combine kinematic and physics estimates for pose/motion estimation, either in the introduction or related works. For example:
- Ugrinovic et. al., “MultiPhys: Multi-Person Physics-aware 3D Motion Estimation” (CVPR 2024).
- Zhang et. al. “PhysPT: Physics-aware Pretrained Transformer for Estimating Human
Dynamics from Monocular Videos” (CVPR 2024).

**Questions:**

- L38: To be clear, what the authors refer to as a "differentiable physics simulation" is the use of rigid body dynamic equations or is it an actually differentiable simulator, e.g.,  Tiny Differentiable Simulator from PyBullet, similar to Gartner et. al?
- L116: Authors create a proxy character (shown in appendix B). I wonder if it is possible to use the humanoid generated by SimPOE or KinPoly for the same end? This latter humanoid seems to be the most realistic representation of a 3D human body for simulation. I would like what the authors think about this and why they chose this specific form.
- Sec. 3.2: How are the contacts modeled? Are they modeled directly with the GRU and data annotations or is there also a specific physics formulation to account for these contacts?

**Limitations:**

Yes.

---

> ### Author Rebuttal · Authors · 2024-08-06
>
> We thank the reviewer for the very positive assessment and the recognition of our method's novelty. We would like to provide answers and clarifications to the remaining questions of the reviewer.
>
> **Presentation Improvement.**
>
> We appreciate the suggestions of the reviewer. We will change the visualization to SMPL model for the GT, kinematics and OSDCap prediction in the camera-ready version of the paper, similar to Figure*2 in the rebuttal PDF document. We are also open to any other recommendations to further improving the presentation quality.
>
> **Additional physics-based metrics.**
>
> We provide additional metrics for physic-based measurements in Table*2 of the rebuttal PDF above. OSDCap helps refining the input kinematics on most of the physics-based metrics. Note that TRACE outperforms our approach in the ground penetration metric. The reason is that in most cases the TRACE predictions float above the ground, which gives a low penetration error but can be seen as equally bad. Thus, we additionally provide a ground distance metric (GD) to reflect the correct foot-ground quality during contact.  The value is computed as the mean absolute vertical differences between foot contact points and ground plane during contact duration, expressed as:
>
> $GD=\frac{1}{6} \sum_{c=1}^{6} \rho_{c} |p_{c}^{OSD} - p_{c}^{GT}|$,
>
> where $\rho_{c}$ is the binary label of contact $c$, $p_{c}^{OSD}$ and $p_{c}^{GT}$ are the 3D vertical positions of contact $c$. There are a total of 6 contact points considered, 3 in each foot accounting for heel, foot and toe.
>
> **Differentiable simulation in OSDCap.**
>
> The differential simulation in line 38 refers to the usage of rigid body dynamics equations and Euler's integration on a proxy character, not an actual physics engines like TDS or PyBullet.
>
> **Selection of a simple proxy character.**
>
> We based the calculation of the inertial properties of the proxy character on the Rigid Body Dynamics Library (RBDL) which was shown to be effective by Shimada et al. [PhysCap]. Since the simpler character definition (as for example compared to KinPoly) yielded good results, we decided to stick with it and mainly focusing on implementing our core contribution of the learnable Kalman filter. During experimentation we found that prediction errors mostly occur from insufficient kinematic or dynamic predictions than from using a more sophisticated proxy character. However, in the future we plan to extend this approach to other external forces, e.g. from full body contact, which will strongly benefit from a detailed human body model such as KinPoly.
>
> **How did foot-ground contacts are modelled?**
>
> The foot-ground contacts are modelled directly by the NN model and automatic data annotation, using the ground truth 3D poses from the training data set.
>
> **Additional baselines.**
>
> Thank you for the additional baseline suggestions, MultiPhys and PhysPT. We will include their results and contributions into the related section. We would like to briefly discuss the suggested paper due to their high relevance:
>
> - MultiPhys introduces a method for multi-person physics-based 3D human motion capture that mainly address plausibility of the captured multi-human poses, inside an physics engine and a reinforcement learning framework. This work could bring new insights about inter-person body interaction, and a more detailed proxy character such as KinPoly (as suggested by the reviewer) is needed.
>
> - PhysPT provides a pre-trained transformer model specifically for human dynamics capture. Instead of explicitly defining physical constraints, as in our or related [NeurPhys, DnD], PhysPT realizes the equations of motion as the main training objective, forcing the transformer model to be physics-aware, thereby allowing a violation of physics laws. The pretrained model can be used to refine kinematics input, acting as an NN-based approximation of a physics simulation.
>
> **Typos.**
>
> Thanks for catching the two typos. We will correct them.

---

> > ### Comment · Reviewer_BVYQ · 2024-08-11
> > **Feedback**
> >
> > Thanks for the authors’ response and the rebuttal document. I have some additional comments/questions.
> >
> > #### **Presentation Improvement.**
> > I see what you did in Figure*2, in my opinion this looks much better. This is more of a nuance, I would further suggest breaking each sequence in two images: one image corresponding to the baseline vs. GT and the other for the proposed method vs. GT, this way the comparison is even clearer. Otherwise you could be masking failures of either the baseline or your method behind the clutter.
> >
> > #### **Physics-based metrics.**
> > The results presented in Table*2 of the rebuttal look good. The use of the GD metric makes sense to me as it is true that predicted motions that contain floating can falsely show very good GP metrics. Also, the results presented are quite good for the proposed method which makes the work potentially more impactful. Will authors release the code so that these results can be reproduced?
> >
> > #### **Differentiable simulation in OSDCap.**
> > I see, thanks for the clarification. This is more of a curious inquiry: I would like to know the authors’ opinion/intuition on how would these fully featured differential simulators work with their approach, if they use it to replace the rigid body dynamics equations and Euler's integration? Could this framework allow for such a change?
> >
> > #### **Selection of proxy characters.**
> > For clarity, I would like to know what do the authors exactly mean by “dynamic predictions” in this context.

---

> > > ### Author Response · Authors · 2024-08-12
> > >
> > > We thank the reviewer for the response and suggestions. We would like to provide the answers to the additional comments below.
> > >
> > > **Presentation.**
> > >
> > > We also think that separating the visualization into two different figures is a better approach to mitigate the cluttering effects. We will change this in the paper by trimming the uninformative background to make space for the two poses.
> > >
> > > **Public code.**
> > >
> > > Yes, the implementation for training and testing will be made publicly available upon acceptance as promised in the paper.
> > >
> > > **Differentiable engines.**
> > >
> > > Integrating an unconstrained physics simulation into our approach is possible. To our experience, it would bring little to no benefits due to the similar rigid body modelling and numerical integration.
> > > However, the most valuable aspect of a physics simulator is the ability to model contacts and object-interaction effectively. These sophisticated contact modeling results in non-differentiable constrained simulation, leading to an intractable gradient for end-to-end learning.
> > >
> > > Promising progress has been made recently to enable differentiability in physics simulations [G, H], and we are currently working towards integrating these newer simulations into physics-based human motion estimation.
> > >
> > > [G] Werling et al., NimblePhysics: Fast and Feature-Complete Differentiable Physics for Articulated Rigid Bodies with Contact, RSS'2021.
> > >
> > > [H] Hu et al., DiffTaichi: Fast and Feature-Complete Differentiable Physics for Articulated Rigid Bodies with Contact, ICLR'2020.
> > >
> > > **Dynamic predictions.**
> > >
> > > In this context, we refer dynamic predictions to the estimates of the joint torques and ground reaction forces.

---

### Official Review · Reviewer_DQhK · 2024-07-12

**Soundness:** 3
**Presentation:** 2
**Contribution:** 3
**Rating:** 5
**Confidence:** 4

**Summary:**

This paper introduces a novel method to estimate 3D human motion from a single camera, aiming for physical plausibility and accuracy. The authors use a differentiable Kalman-filtering approach in an online setting that balances kinematics estimated from an off-the-shelf algorithm (TRACE) and the physics simulation. They built OSDNet, which consists of fully connected and GRU layers, for the given optimal state human dynamics capture task. They show their method outperforms the existing methods in the same online estimation setting while each technical component was validated through ablation studies.

In general, the method sounds reasonable and novel. However, there are several things that need to be cleared.

**Strengths:**

They propose the approach inspired by the neural Kalman filtering approach. This method seems to be novel.

Compared to their baseline approach (TRACE), they provides significant performance improvement, which is not the case for the other physics-based approaches (if I remember correctly).

**Weaknesses:**

It is not clear how they constructed the character. They mention that they used metadata of Human3.6M dataset. Does it mean they use the ground-truth bone lengths of the subject?
– If so, they should state their method with automatic character generation (similar to what SimPoE did) for a fair comparison. It is not clear if their outperformance is coming from the known limb length assumption or the method itself.
– If not, please state more clearly how they generate the character in the physics simulator.

It is not clear why they selected TRACE as a default method. From Table 1, TRACE seems to significantly underperform VIBE, which has been widely used as a baseline kinematics estimator (e.g., SimPoE). The major advantages of using TRACE are its capability to track a person's identity and the robustness of the camera motion. However, all training and evaluation were done in a fixed camera coordinate system.

In the paper, they mention that since TRACE's output is expressed in the first frame global coordinate, their method is agnostic to the initial calibration (line 473). I disagree with this line. Isn't the initial pose of camera essential to know the gravitational direction? Will this method be robust to the camera which is highly tilted?

In Table 3, conventional PD methods are showing catastrophic performance. I am doubtful about this, maybe the training configuration was not set properly? This is telling us that neural PD control is destroying the performance which does not align with existing literature.

Many of the state-of-the-art video-based kinematics estimation models are not stated in the paper including HybrIK (CVPR 2021), CLIFF (ECCV 2022), PMCE (ICCV 2023), ReFit (ICCV 2023), MotionBERT (ICCV 2023), and HMR2.0 (ICCV 2023). To the best of my knowledge, all these methods (not limited to) have better performance on Human3.6M dataset.

**Questions:**

[Some are duplicated with the Weakness section]
– How did you construct a physics-based 3D person model?
– Why did you use TRACE? IF you value the potential of future use case with moving camera, this needs to be stated.
– Why do neural PD controllers perform so badly in your test case?
– I guess the GRU unit is uni-directional, am I correct?

**Limitations:**

–

---

> ### Author Rebuttal · Authors · 2024-08-06
>
> We thank the reviewer for the feedback and constructive review. We would like to address questions and clarify the concerns below.
>
> **Proxy character creation.**
>
> For character construction, we used the Human 3.6M metadata skeleton as the initialization of our character. The bone lengths are treated as extra learnable parameters and optimized along with the model during the training process. During testing, the bone lengths are fixed to the ones learned during training, i.e. no ground truth bone lengths are used when testing.
>
> **Why TRACE?**
>
> The reason for choosing TRACE as our kinematic estimation input is their additional global translation estimation (w.r.t the first frame), while for example VIBE only predicts root relative poses. We use this global information to enable the integration of physics laws into the system. Moreover, TRACE is a well-established and new (CVPR'2023) baseline. The additional benefit of TRACE being robust to camera motion is an advantage we plan to explore in future work.
>
> **OSDCap estimation is agnostic to initial calibration.**
>
> The statement in line 473 explains how we preprocess the data. It indeed only removes the translational component of the camera calibration. As correctly stated in the review, the rotational component is important for the physics simulation. OSDCap will still work in the case of highly tilted camera, as long as the camera pose is provided. If a fully automatic calibration process for tilted cameras is desired, an automatic calibration with humans as calibration targets can be employed [B].
>
> [B] Tang et al., CasCalib: Cascaded Calibration for Motion Capture from Sparse Unsynchronized Cameras, 3DV, 2024.
>
> **The performance of the PD-only method.**
>
> We thank the reviewer for pointing out this potentially misleading table.
>
> The PD implementations in Table 3 are different from other PD controller-based works (PhysCap, NeurPhys, DnD). While NeurPhys (cf. their implementation on Github) assumes that gravitational forces and external forces are cancelling out, which is not true in the real world, we decide to keep them to maintain physical plausibility. Moreover, to make the PD controllers work, PhysCap, NeurPhys, and DnD add an additional predicted offset term to the controller output which introduces another source of implausibility. By contrast, we maintain the physical plausibility of the simulation and the PD controller which leads to state-of-the-art performance among physics-based approaches. We will clarify this in the final version.
>
> **Additional baselines.**
>
> We will add the suggested kinematics estimation methods in the comparison. However, similar to what have been done in related physics-based methods such as NeurPhys, DiffPhy or DnD, it is not trivial to directly compare physics-based motion reconstruction to image-based pose estimators, since the latter do not predict the global 3D trajectory in world coordinate, making it impossible to evaluate the global motion quality (MPJPE-G, GRP). We kindly refer to the comments to all reviewers above for a detailed discussion.
>
> **Is GRU uni-directional?**
>
> Yes, the GRU unit is uni-directional.

---

> > ### Comment · Reviewer_DQhK · 2024-08-11
> > **Thank you for your rebuttal**
> >
> > Dear Authors,
> >
> > Thank you so much for your solid rebuttal on my review. After carefully reading your response, I would like to increase my initial score to "5".

---

> > > ### Author Response · Authors · 2024-08-12
> > >
> > > We thank the reviewer for the assessment and the upgrade of the rating.

---

### Official Review · Reviewer_wqMZ · 2024-07-15

**Soundness:** 3
**Presentation:** 3
**Contribution:** 3
**Rating:** 5
**Confidence:** 4

**Summary:**

The kinematic motion estimation suffers from inconsistencies of frame-wise predictions while the physics-based method suffers from the gap between the simulator environment and the real-world ground truth. This paper proposes a method to taking advantages of both by connecting them by a learnable kalman filter network. The proposed method OSDCap is a new physics-based human motion and dynamics estimation method. The proposed Kalman filter takes the simulated motions and the 55 noisy 3D pose estimation as inputs, combines them, and produces an optimal state prediction as the 56 output.  It also has a learnable inertia prediction for weights distribution, that produces plausible motion as well as valuable estimates of exterior forces and internal torques.

**Strengths:**

- The proposed method is very straightforward and intuitive. Considering the noise pattern of the frame-wise kinematic method (frame independent) and the noise of physics-based method (temporal propagated), the idea matches with the design motivation of kalman filter very well.
- The presentation is good. I could understand the high-level idea and the technical design without trouble.

**Weaknesses:**

- Some related works are missing in the experiments, especially considering that kalman filter is widely used for human pose tracking. Moreover, as the authors proposed a learnable module to replace the role of Kalman filter (by predicting the gains), it would be helpful to compare the proposed method and a baseline with the classic kalman filter parametric model.
- There are some missing baselines, especially the kinematic ones in the Table 1, for example PointHMR (CVPR’2023) and KTPFormer (CVPR2024), considering that the papers are available before the submission of this work, is there any reason that some recent works are not listed in the comparison?

**Questions:**

I am overall satisfied with the novelty of the proposed method with no previous work having implemented the straightforward but well-connected idea as presented in this work as far as I know. However, I will need more evidence about the experimental significance to adjust my rating. Or, could the authors clarify the reason of the missing of more recent baseline works? I would need more evidence to recognize that the kalman filter (at least the learnable one in this work does work generalizability  and is more superior than the classic parametric KF).

**Limitations:**

The authors have discussed the limitations in the paper. They use the widely distributed datasets with human faces but they are controlled in the studio environment and under consent.

---

> ### Author Rebuttal · Authors · 2024-08-06
>
> We thank the reviewer for the positive assessment to our proposed idea and writing quality. For a better interpretation of our work, we clarify and answer the questions of the reviewer below.
>
> **Comparison to a classical Kalman Filter.**
>
> Classical Kalman filters have been used traditionally for all types of tracking problems, including human pose tracking. More recently, the only work we could find that utilizes a Kalman filter is from Buizza et al. [A], where a classical parametric Kalman filter is employed to track 2D keypoints estimated from an off-the-shelf 2D human pose estimators. We are happy to take further suggestions for related work that we might have missed.
>
> We agree that our learnable Kalman filter should be compared to a classical Kalman filter. Therefore, we conducted an additional experiment where we replace our learnable filter by a traditional one. The biggest challenge of using classical Kalman filter is the tuning of unknown noise covariances of both the kinematic input TRACE and the simulated result from PD controller. Assuming noise covariances that are constant over time and equal in all directions, the ratio between the noise covariance of the simulated PD controller (process noise) and the noise covariance of the kinematic input TRACE (measurement noise) governs the quality of the Kalman filter estimates. The evaluation results can be seen in Table*1 in the PDF file, where we use constant noise covariances with ratios 100/1, 10/1, 1/1, 1/10, 1/100 between process noise and measurement noise. While a classical KF approach helps increase the result marginally, optimal results are difficult to find. Our choice of a learnable Kalman filter relieves us from trial-and-error process of finding the correct noise covariance matrices and achieves the best results.
>
> [A] Buizza et al., Real-Time Multi-Person Pose Tracking using Data Assimilation, WACV, 2020.
>
> **Additional baselines.**
>
> We will add the suggested baselines and others suggested by reviewer DQhK to the paper. Please refer to our global rebuttal response for references about their performances. We would like to briefly discuss the two suggested approaches in relation to our work.
>
> - PointHMR (CVPR 2023) is a frame-wise mesh and keypoint regression method that can work in an online setting. However, unlike TRACE, PointHMR does not produce global root translation of the estimated poses, therefore, can only be used for MPJPE reference, not for physics-based reconstruction methods.
>
> - KTPFormer (CVPR 2024) predicts 3D keypoints directly instead of 3D joint angles. This might lead to implausibilities in the skeletal configuration, for example in changing bone lengths, and is therefore not transferable to a dynamics model. Since it appears to be the current state of the art in terms of the common metrics like MPJPE we will include it.

---

> > ### Comment · Reviewer_wqMZ · 2024-08-13
> >
> > I appreciate the efforts by the authors to address my concerns.
> >
> > I agree that the hyperparameter setting in the classic Kalman filter formulation can be tricky to adjust especially when there are combined factors to consider. Considering the covariance for kinematic input and PD controller outcomes, Multivariate Kalman filtering can be a parametric solution that may be worth future study. Kalman filtering family has been studied for decades but it is still one of the go-to solution in modern tracking and other time series analysis tasks. A comprehensive comparison with different settings and members of this family can significantly improve the convincingness of this paper's soundness.
> >
> > I am in general satisfied with the responses from the authors and maintain my positive rating towards the acceptance of this paper.

---

> > > ### Author Response · Authors · 2024-08-13
> > >
> > > We thank the reviewer for the feedback and final evaluation.
> > >
> > > We performed a smaller study in on different parametrizations for a standard Kalman filter in the response to all reviewers above. A more comprehensive study on different variances of Kalman filter family on human motion capture will be investigated as a natural progression of the paper. We are happy to include another parametrization if desired.

---

> > > > ### Comment · Reviewer_wqMZ · 2024-08-13
> > > >
> > > > Thanks for the promise from the authors.
> > > >
> > > > I have seen many papers in the past years claiming to use learnable motion modules to beat classic linear filtering-based motion modeling for tracking or forecasting. However, after investigating the new methods on different data patterns and removing the noise from post-processing tricks, etc, I seldom find methods capable of consistently beating the Kalman filter in the area of visual tracking. This is why the most widely adopted multi-target tracking arts in practice is still a powerful detector + a Kalman filter-based tracker, such as SORT[1]/OC-SORT[2]/ByteTrack[3] even in 2024.
> > > >
> > > > I will be looking forward to the investigation from the authors.
> > > >
> > > > > [1]: "Simple Online and Realtime Tracking"
> > > >
> > > > > [2]: "Rethinking SORT for Robust Multi-Object Tracking"
> > > >
> > > > > [3]: "ByteTrack: Multi-Object Tracking by Associating Every Detection Box"

---

> > > > > ### Author Response · Authors · 2024-08-14
> > > > >
> > > > > We thank the reviewer for the additional references and the productive discussion.
> > > > >
> > > > > Kalman filtering prevails due to its simple yet effective design in tracking problems. The field of tracking is evolving, new challenges and environmental setups have been posed, i.e. 3D human motion tracking from videos. Therefore, it is worthwhile for deeper investigations of Kalman filter and its newer variances to address those problems optimally in real-world applications.

---

### Author Rebuttal · Authors · 2024-08-06

We would like to express our gratitude towards the reviewers for their helpful reviews and assessment!

We are happy that the reviewers recognize that the proposed method OSDCap for physics-based 3D human motion capture is novel (wqMZ, DQhK, BVYQ, QEqx), of high significance to the field (BVYQ), physically explainable (QEqx), and fully differentiable (mMU1). Reviewer BVYQ comments that our paper is the first to successfully combine kinematics estimation with differentiable simulation in and end-to-end manner, providing a promising direction to bridge the gap between kinematics and physics. Three reviewers (wqMZ, BVYQ, QEqx) agree that the manuscript is well-written and easy to follow.
In this section, we provide the answers to the common question asked by multiple reviewers. Further on, we address the reviewers' questions individually in the respective sections.

**Additional references.**

Reviewers wqMZ, DQhK and BVYQ pointed out that a comparison to other human pose estimation work is missing. We followed the common practice in the field of physics-based human motion estimation (DiffPhy, NeurPhys, DnD) and compared only to other physics-based approaches in our main experiments, since their objective of creating a plausible motion is the same as ours. However, we agree that for a better interpretation of the performance of physics-based approaches, including traditional approaches for comparison is beneficial and we provide results in the following table for the Human 3.6M dataset.

| Method | Venue | MPJPE | MPJPE-PA |
|:----------------|:----------------:|:----------------:|:-----------------:|
| HybrIK | CVPR'2021 | 54.4 | 34.5 |
| PMCE | ICCV'2023 | 53.5 | 37.7 |
| PhysPT | CVPR'2024 | 52.7 | 36.7 |
| ReFit | ICCV'2023 | 48.5 | 32.4 |
| PointHMR | CVPR’2023 | 48.3 | 32.9 |
| CLIFF | ECCV'2022 | 47.1 | 32.7 |
| MotionBERT | ICCV'2023 | 43.1 | 27.8 |
| KTPFormer | CVPR'2024 | 33.0 | 26.2 |

However, while these produce a low MPJPE, most of them do not estimate the global motion and more importantly might not be physically plausible, including strong jitter and floating/ground penetration. By contrast, although our approach sometimes produces worse distance-based metrics (MPJPE, PCK, etc.), it produces significantly more plausible motions as well as mitigating unnatural floating or ground penetration. We think that the novel concept of a learnable Kalman filter that produces state-of-the-art results for physics-based pose estimation outweighs the pure distance-based evaluation of other approaches.

**The accompanying PDF document.**

To fully address reviewers' questions and concerns, we conduct some additional measurements on the output of the proposed method OSDCap. The PDF contains:

- Table*1, evaluation results of using a classical Kalman Filter to combine the physics simulation and the kinematics observation. This table is relevant to our response to Reviewer wqMZ.
- Table*2, additional physics-based metrics of the estimated human motion from OSDCap. The metrics consist of ground penetration (GP), friction loss, velocity loss, and foot skating. Since the GP metric cannot correctly reflect the floating artifact of the estimated pose, we also compute a ground distance metric (GD) to measure the foot-ground contact quality. This table is to address the concerns from Reviewers BVYQ and Reviewer mMU1.
- Figure*1, the overlay re-projected human pose, presented as SMPL model, onto the input 2D images. This figure is in response to Reviewer mMU1 about discrepancy evaluation.
- Figure*2. the 3D visualization of OSDCap's estimated pose, input kinematics TRACE, and the ground-truth pose provided by the Human 3.6M dataset. The figure is created according to the suggestions of Reviewer BVYQ for a better presentation of the results.

Above is the common information that is relevant to our individual responses to all reviewers. We further address each reviewer's concerns respectively in the below sections.

---

### Decision · Program_Chairs · 2024-09-25

**Decision:**

Accept (poster)

**Comment:**

The paper proposes an approach to estimate the 3D pose of a single-person from a monocular camera image. The approach uses a differentiable Kalman-filter that combines a kinematic motion estimator a differentiable physics formulation, which can be trained using a supervised learning approach. Doing so, the approach addresses limitations of previous work (such as jittering). Moreover, the approach outperforms the existing methods in the same online estimation setting while each technical component was validated through ablation studies.

The reviewers mainly raised issues considering related work and baselines, as well as technical obscurities. However, the authors were able to clarify the open points and provide the missing quantitative results during the rebuttal period.

Decision:
After a fruitful discussion all reviewers voted for acceptance. I therefore recommend to accept the paper and encourage the authors to use the feedback provided to improve the paper for its final version.